# A Structure-Aware Generative Adversarial Network for Bilingual Lexicon Induction

**Bocheng Han [1], Qian Tao[1]**[*]**, Lusi Li[2],** and **Zhihao Xiong[3]**

[1]South China University of Technology, Guangzhou, China
[2]Old Dominion University, Norfolk, USA
[3]Baidu Inc., Beijing, China
`202221045875@mail.scut.edu.cn`, `taoqian@scut.edu.cn`
`lusili@cs.odu.edu`, `xiongzhihao@baidu.com`

## Abstract

Bilingual lexicon induction (BLI) is the task of inducing word translations with a learned mapping function that aligns monolingual word embedding spaces in two different languages. However, most previous methods treat word embeddings as isolated entities and fail to jointly consider both the intra-space and inter-space topological relations between words. This limitation makes it challenging to align words from embedding spaces with distinct topological structures, especially when the assumption of isomorphism may not hold. To this end, we propose a novel approach called the Structure-Aware Generative Adversarial Network (SA-GAN) model to explicitly capture multiple topological structure information to achieve accurate BLI. Our model first incorporates two lightweight graph convolutional networks (GCNs) to leverage intra-space topological correlations between words for generating source and target embeddings. We then employ a GAN model to explore inter-space topological structures by learning a global mapping function that initially maps the source embeddings to the target embedding space. To further align the coarse-grained structures, we develop a pairwise local mapping (PLM) strategy that enables word-specific transformations in an unsupervised manner. Extensive experiments conducted on public datasets, including languages with both distant and close etymological relationships, demonstrate the effectiveness of our proposed SA-GAN model.[1]

## 1 Introduction

Bilingual lexicon induction (BLI) has emerged as a crucial task in natural language processing (NLP), focusing on the discovery of corresponding words between two languages using monolingual corpora. Due to its ability to facilitate the transfer of semantic knowledge between languages, BLI has been successfully applied in various NLP applications, including machine translation (Artetxe et al., 2018c; Ren et al., 2020), cross-lingual sentiment analysis (Singh and Lefever, 2020) and text classification (Dong and de Melo, 2019).

Most BLI methods aim to learn a mapping function that aligns word embeddings of two languages into a shared embedding space, which allows leveraging independently trained monolingual embeddings and then utilizing the learned mapping to generate bilingual lexicons (Mikolov et al., 2013; Glavaš et al., 2019). Thereinto, Mikolov et al. (2013) first observed that a linear orthogonal mapping proved to be empirically effective in transforming the source embedding space to the target language's space. This mapping was achieved by minimizing the squared Euclidean distance between the translation pairs in a given parallel vocabulary. They attribute the success of their method to the isomorphic assumption that the two embedding spaces exhibit similar geometric structures as they found that the linear projection outperformed its non-linear counterpart with multilayer neural networks. Building upon this work, various BLI methods have been proposed to improve the inductive performance by enforcing an orthogonality constraint (Lample et al., 2018), normalizing the embeddings (Artetxe et al., 2018a), relaxing the isomorphic assumption (Patra et al., 2019), leveraging the clique-level information (Ren et al., 2020), refining with Coherent Point Drift algorithm (Cao and Zhao, 2018; Oprea et al., 2022), distinguishing the relative orders (Tian et al., 2022), etc. From them, it can be noticed that reliable mapping functions can be learned even with weak supervision.

Furthermore, recent advancements have introduced several unsupervised models through adversarial training to learn mapping functions without the need for parallel data (Lample et al., 2018; Bai et al., 2019; Mohiuddin and Joty, 2019; Xiong and Tao, 2021), offering a data-driven, scal-

---

[*]Corresponding author.
[1]Our code is available on `https://github.com/scutBCH/SAGAN`

able, and language-independent approach to induce cross-lingual representations from low-resource languages. However, existing adversarial methods focus on word-level alignment and treat the words in the embedding space as isolated entities, ignoring the underlying topological structures among words. Therefore, the relationship between words is not preserved and the topological structure of the embedding spaces is not well exploited or involved during training, leading to poor performance compared with other non-adversarial methods (Artetxe et al., 2018b; Ren et al., 2020).

In addition, conventional BLI methods typically assume that the embedding spaces of different languages are nearly isomorphic, and they learn a global linear mapping function shared by all words based on this assumption. However, recent studies (Søgaard et al., 2018; Patra et al., 2019) have found that the isomorphic assumption may not hold strictly due to deviations in the distributions of word embeddings for different languages. Consequently, the performance of BLI methods might be degraded, especially for the language pairs far from isometry. In such a case, a globally-aligned mapping function may not be an optimal solution. There have been some approaches that attempted to alleviate this problem by learning personalized mapping functions for different words or employing supervised non-linear mapping in latent space (Glavaš and Vulić, 2020; Tian et al., 2022; Mohiuddin et al., 2020). However, the supervised signals are all indispensable to their proposals and cannot be applied to the unsupervised learning setting without any labeled data.

To address these challenges, we propose a novel unsupervised model called structure-aware generative adversarial network (SA-GAN) to explicitly capture multiple topological structure information for accurate BLI. Specifically, given a source language and a target language, SA-GAN first views the embedding space of each language as a graph and utilizes two lightweight graph convolutional networks (GCNs) to encode two embeddings for exploring the intra-space topological structures. With the extracted structural information, we formulate the learning of a mapping function in a fashion that admits an adversarial game. SA-GAN employs a GAN model to learn a linear mapping matrix, allowing for the global mapping of the extracted source embeddings into the target embedding space. Unlike previous adversarial methods that usually enforce an orthogonality constraint on the mapping function, SA-GAN removes this constraint during adversarial training since the isomorphic assumption may not hold true practically. The learned mapping matrix facilitates the construction of a seed dictionary. To further refine the coarse-grained structures and enhance the seed dictionary, SA-GAN introduces a pairwise local mapping algorithm (PLM). This algorithm can learn word-specific transformations for different words based on their nearest neighbors within the seed dictionary. By doing so, our method reduces reliance on isometry and achieves improved BLI performance in a fully unsupervised manner. To verify the effectiveness of SA-GAN, we conduct extensive experiments with sixteen different language pairs, comprising both etymologically distant and close languages to thoroughly test our model performance with varying degrees of isomorphism between monolingual spaces. Experimental results show that our model can achieve comparable performance to state-of-the-art unsupervised methods in most cases and even surpass previous supervised ones. Our main contributions can be summarized as follows:

- We develop a novel adversarial framework SA-GAN to explore both the intra-space and inter-space topological information for unsupervised BLI. It integrates two GCNs and a GAN to learn a linear mapping function through adversarial training without imposing an orthogonality constraint, providing greater flexibility in aligning different languages where the isomorphic assumption may not hold.

- We propose a pairwise local mapping (PLM) algorithm, which enables the learning of word-specific transformations. PLM utilizes topological information from the nearest neighbors in the seed dictionary to refine the alignments and alleviate the reliance on isometry.

- We conduct extensive experiments over popular benchmarks, and the results demonstrate that our model outperforms existing unsupervised methods and even outperforms supervised state-of-the-art methods.

## 2   Methdology

In this paper, we denote the source and target language word embeddings as $\mathbf{X} \in \mathbb{R}^{d \times n}$ and

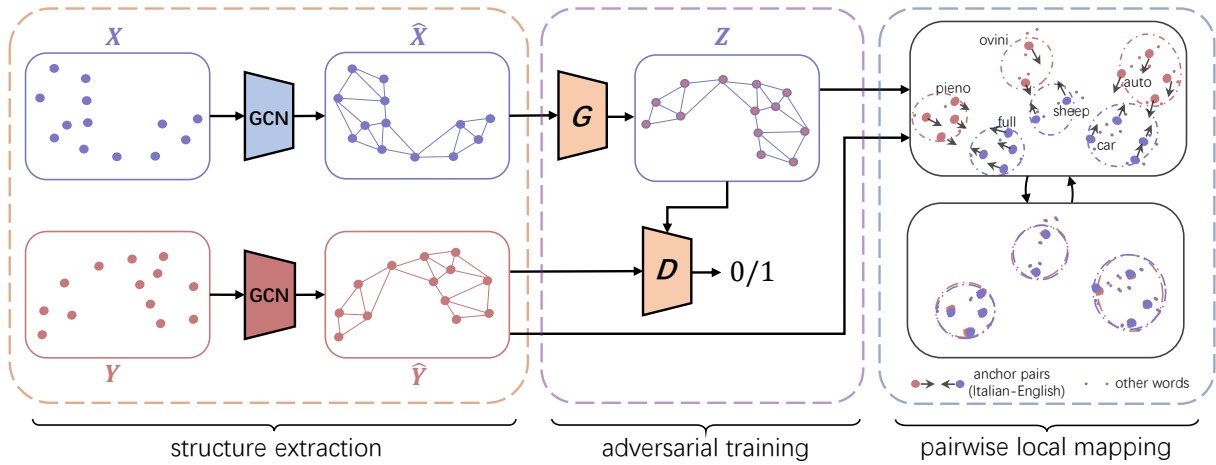

Figure 1: An overview of our proposed SA-GAN framework.

$\mathbf{Y} \in \mathbb{R}^{d \times m}$, where $n$ and $m$ are the numbers of words in $X$ and $Y$, respectively, and $d$ stands for the embedding size. Our proposed SA-GAN method consists of three major components, including structure extraction, adversarial training, and pairwise local mapping, as shown in Figure 1. Each module has its own role to play while targeting different goals. By the strategy of splitting, each module can focus more on its task and improve the overall performance while reducing the complexity. Specifically, given two monolingual embeddings $\mathbf{X}$ and $\mathbf{Y}$, we first capture the topological information of each language via two lightweight GCN modules. After that, a global mapping matrix is learned via adversarial training, which transforms the source word embeddings into the target embedding space. Finally, SA-GAN designs a novel PLM algorithm to learn word-specific transformation, which alleviates the reliance on isometry. We will next formally introduce the model.

## 2.1 Structure Extraction

Recently, graph neural networks (GNNs) have been widely utilized in various fields due to their powerful ability to extract spatial information from graphs. Inspired by this, we propose to incorporate a GNN module prior to adversarial training to exploit the topological correlations in the embedding spaces by viewing the entire embedding space as a graph. In this graph, each word is represented as a node and edges connect it to its $k$-nearest neighbors. The graph can be denoted as $G = (V, E, A)$, where $V = \{v_1, ..., v_n\}$ represents $n$ nodes and $n$ is the total number of vocabulary words in one language; $E = \{e_{i,j}\}_{i,j=1}^{n}$ is a set of edges, where each edge $e_{i,j}$ is associated with a weight $A_{i,j}$ in adjacency matrix $A$ to describe the similarities between the words $v_i$ and $v_j$ in the graph.

$$A_{i,j} = \begin{cases} -\ln \|\mathbf{x}_i - \mathbf{x}_j\|_2, & i \neq j, \\ 0, & i = j. \end{cases} \quad (1)$$

where $\mathbf{x}_i$ and $\mathbf{x}_j$ are the word embeddings for node $v_i$ and node $v_j$, respectively.

The basic idea of a GNN is to learn node representations in a graph by incorporating information from neighboring nodes through iterative aggregation and transformation processes. During the aggregation process, the neighboring node features are aggregated to generate a combined representation for each node. During the transformation process, the combined representations undergo a transformation to generate refined node representations using neural network layers for capturing more complex topological relationships. A well-known example of a traditional GNN is the Graph Convolutional Network (GCN) (Kipf and Welling, 2017). GCN leverages convolutional layers on the graph structure to perform neighborhood aggregation and transformation as follows.

$$X^{(l)} = \sigma(\hat{A} X^{(l-1)} W^{(l)}) \quad (2)$$

$$\hat{A} = D^{-\frac{1}{2}} (A + I) D^{-\frac{1}{2}} \quad (3)$$

where $X^{(l)}$ and $X^{(l-1)}$ denote the embedding representations after $l$ and $(l-1)$ layers propagation for all the $n$ nodes and $X^{(0)} = X$; $\hat{A}$ is the normalized and regularized adjacency matrix; $I$ is an identity matrix, which is added on $A$ to include self-connections; $D$ is a diagonal node-degree matrix. $W^{(l)}$ is the feature transformation matrix at the $l$-th layer and $\sigma(\cdot)$ is an activation function. However,

the full gradient descent strategy is often used to train GCN, suffering from high computational complexity for large-scale datasets. Hence it is difficult to fit in with subsequent adversarial training, where mini-batch stochastic gradient descent (SGD) is used for each update. Some researchers (Hamilton et al., 2017) have proposed mini-batch SGD for GCN to alleviate the problem, but the overheads of these methods are still large.

Motivated by He et al. (2020), we propose a simplified GCN by removing the activation function $\sigma(\cdot)$ and the feature transformation matrices $\{W\}_{l=1}^{L}$, defined as follows:

$$X^{(l)} = \hat{A}X^{(l-1)} \qquad (4)$$

$$\hat{A} = D^{-1}(A + I) \qquad (5)$$

Furthermore, to reduce the computation time, we construct a $K_g$-nearest graph to preserve the edge connections of the top $K_g$ nearest neighbors for each node and keep the adjacency matrix $A$ as a sparse matrix. Lastly, we combine the embeddings obtained at each layer to produce the final embedding matrix:

$$\hat{X} = \alpha_0 X^{(0)} + \alpha_1 X^{(1)} + ... + \alpha_L X^{(L)} \qquad (6)$$

where $\{\alpha_l\}_{l=0}^{L}$ are the tradeoff coefficients.

It is worth noting that there are no trainable parameters of the designed GCN module. In other words, rather than training the propagation process at each iteration, the final embedding matrix only needs to be precomputed once and can be stored as a constant, which greatly decreases the computational cost and memory resource requirements.

Two GCN modules are respectively applied to the source language $\mathbf{X}$ and the target languages $\mathbf{Y}$ to form the new embedding representations $\hat{\mathbf{X}}$ with $n$ nodes and $\hat{\mathbf{Y}}$ with $m$ nodes, which contain the topological structure information of the source and target embedding spaces.

## 2.2 Adversarial Training

With the extracted embedding representations, our goal is to match them for inducing a seed dictionary. Recent studies have demonstrated the effectiveness of adversarial training in aligning two distributions (Lample et al., 2018; Xiong and Tao, 2021). Building upon this concept, we employ adversarial training through a GAN in our work to learn a mapping function in a fully unsupervised manner.

Specifically, we train a generator $G$ to learn a linear mapping matrix $W$ to deceive a discriminator $D$. The generator $G$ aims to map the word embeddings from the source language to the target language through $G(\hat{x}_i) = W\hat{x}_i$. $G$ can be trained with the loss function as follows:

$$\mathcal{L}_{G|D} = -\frac{1}{n} \sum_{\hat{x}_i \in \hat{X}} \log(D(W\hat{x}_i)) \qquad (7)$$

The discriminator $D$ is trained to distinguish between the mapped source embeddings $W\hat{X} = \{W\hat{x}_1, ..., W\hat{x}_n\}$ and the target embeddings $\hat{Y} = \{\hat{y}_1, ..., \hat{y}_m\}$ using the cross-entropy loss:

$$\mathcal{L}_{D|G} = -\frac{1}{m} \sum_{\hat{y}_i \in \hat{Y}} \log D(\hat{y}_i) \\ -\frac{1}{n} \sum_{\hat{x}_i \in \hat{X}} \log(1 - D(W\hat{x}_i)) \qquad (8)$$

At each iteration, we optimize the generator loss (Equation (7)) and the discriminator loss (Equation (8)) alternately with stochastic gradient updates. Through adversarial training, we can obtain an initial solution of $W$. Following other GAN-based methods (Lample et al., 2018; Bai et al., 2019; Xiong and Tao, 2021), we further refine the learned mapping $W$ via a self-learning strategy in (Artetxe et al., 2018b) by iteratively solving the *Procrustes* problem and applying a dictionary induction step. In our self-learning, we run five iterations of this process.

Although the word embeddings $\hat{\mathbf{X}}$ and $\hat{\mathbf{Y}}$ contain the structure information using GCNs, they also introduce a challenge known as over-smoothing (Li et al., 2018). This issue arises when the words become indistinguishable from each other, especially those words lying in dense areas, leading to poorer performance when inducing the bilingual lexicon. To address this concern, we utilize $\hat{\mathbf{X}}$ and $\hat{\mathbf{Y}}$ for finding the initial solution $W$. Subsequently, we discard $\hat{\mathbf{X}}$ and $\hat{\mathbf{Y}}$, and the remaining processes, including self-learning and the PLM algorithm (Section 2.3), are executed using the original embeddings $\mathbf{X}$ and $\mathbf{Y}$. This decision is made to mitigate the over-smoothing problem and ensure that subsequent steps operate on the unaltered embeddings, thus potentially improving the performance of bilingual lexicon induction.

## 2.3 Pairwise Local Mapping Algorithm

With structure extraction and adversarial training, we can capture valuable structural information and

learn a mapping function that is shared globally by all words under the isomorphic assumption. However, several studies (Ruder et al., 2019; Patra et al., 2019) have found that this assumption is not strictly applicable, and it may lead to poor performance in BLI, particularly for language pairs that deviate significantly from isometry. In this situation, a global-shared mapping function may not be the optimal solution. To further refine the alignments, we introduce a novel PLM algorithm to recompute and upgrade the embedding representations for different words based on the learned seed dictionary and improve the BLI performance.

Our PLM algorithm consists of two steps: generating a seed dictionary $D(\mathbf{Z}^D, \mathbf{Y}^D)$ and then utilizing the word pairs in this synthetic dictionary to perform a local mapping for each word. Firstly, we induce a seed dictionary utilizing the learned mapping matrix $W$ to map the source word embeddings to the target embedding space as follows:

$$\mathbf{Z} = W\mathbf{X} \quad (9)$$

where $\mathbf{Z}$ is the mapped source word representations. With $Z$ and $Y$, we can retrieve the translation pairs and build the seed dictionary $D(\mathbf{Z}^D, \mathbf{Y}^D)$ according to the cross-domain similarity local scaling (CSLS) measurement (Lample et al., 2018). Specifically, given a mapped source word $\mathbf{z}$, we treat the nearest word in the target embedding space as the translation results as

$$\mathrm{CSLS}(\mathbf{z}, \mathbf{y}) = 2\cos(\mathbf{z}, \mathbf{y}) - r_T(\mathbf{z}) - r_S(\mathbf{y}) \quad (10)$$

where $r_T(\mathbf{z})$ is the average cosine similarity between $\mathbf{z}$ and its $k$-nearest neighbors in $Y$; $r_S(\mathbf{y})$ is the average cosine similarity between $\mathbf{y}$ and its $k$-nearest neighbors in $Z$. To refine the quality of the dictionary, we filter out word pairs in the generated dictionary that are not the $K_m$ most frequency words in each language which are usually of low quality and induce word pairs from both directions in the seed dictionary $D(\mathbf{Z}^D, \mathbf{Y}^D)$.

Secondly, we use the word pairs in this synthetic dictionary to improve the mapped embedding and get a pair-wise local mapping for each word. Given a mapped source word $\mathbf{z}_i$, we first obtain its top $K_a$-nearest neighbor words $\mathbf{z}_1^D, ..., \mathbf{z}_{K_a}^D$ from $\mathbf{Z}^D$ as anchors, denoted as $\mathcal{N}_i$ with a coefficient for each anchor point:

$$e_{ij} = \cos(\mathbf{z}_i, \mathbf{z}_j^D) \quad (11)$$

that indicates the importance of anchor word $\mathbf{z}_j^D$ in $\mathcal{N}_i$ to the given source word $\mathbf{z}_i$. The closer an anchor is to $\mathbf{z}_i$, the larger the importance coefficient it gets. However, since the cosine similarity ranges from 0 and 1, we observe that even the anchors that are too far to give a useful guideline still get a high coefficient, ie. 0.4, which will introduce potential noises to the pairwise mapping. To avoid the influence, we scale the importance using the softmax function with temperature $\tau$:

$$a_{ij} = \mathrm{softmax}_j(e_{ij}) = \frac{\exp(e_{ij}/\tau)}{\sum_{k\in\mathcal{N}_i} \exp(e_{ik}/\tau)} \quad (12)$$

which increases the influence of the nearest neighbor anchors even further and decreases for the distant ones. We then compute the new embedding representation of $\mathbf{z}_i$ with the guidelines of generated dictionary, as follows:

$$\mathrm{PLM}(\mathbf{z}_i) = \mathbf{z}_i + p \cdot \sum_{k\in\mathcal{N}_i} a_{ik} \cdot (\mathbf{y}_k^D - \mathbf{z}_k^D) \quad (13)$$

where $p$ is the rate for updating the word embeddings.

The above steps can be iteratively done for both directions and at each iteration, we regenerate the dictionary $D$ with the updated embedding representation in the previous iteration to further improve the quality of the synthetic dictionary.

## 2.4 Training Paradigm

In summary, the proposed approach first extracts the structure information using GCNs and learns a global mapping function in an adversarial manner to map the embeddings of two languages into the same space. In order to alleviate reliance on isometry, we further apply the PLM algorithm to learn pairwise mapping functions for different words based on the learned seed dictionary. The whole training process of the proposed approach is unsupervised and described in Algorithm 1.

## 3 Experiment

### 3.1 Experimental Settings

Following the common practice of BLI, we evaluate the performance of induced language pairs with the Precision at 1 (Precision@1) metric, which measures the word translation accuracy in comparison to a gold standard.

**Dataset** To demonstrate the effectiveness of our SA-GAN model, we leverage the widely used

**Algorithm 1:** Training procedure of model

---

**Data:** Normalized monolingual word embeddings **X** for source language and **Y** for target language

1  Build adjacency matrix $A$ according to Eq.1;
2  Extract structural information following Eq.4 and get new embedding representation $\hat{\mathbf{X}}$ and $\hat{\mathbf{Y}}$;
   /* Learn global shared transformation */
3  **for** *n_epochs* **do**
4    **for** *n_iterations* **do**
5      **for** *k steps* **do**
6        Sample a batch from $\hat{\mathbf{X}}$ and $\hat{\mathbf{Y}}$ ;
7        Update discriminator with Eq.8;
8      **end**
9      Sample a batch from $\hat{\mathbf{X}}$ and $\hat{\mathbf{Y}}$ ;
10     Update generator $W$ on adversarial loss to fool discriminator with Eq.7;
11   **end**
12   Use validation criterion to save the best model;
13 **end**
14 Refine the learned mapping $W$ via self-learning process;
15 Map the source embeddings **x** to the target embeddings space: $\mathbf{Z} \leftarrow W\mathbf{X}$;
   /* Learn word-specific transformation */
16 **for** *m_iterations* **do**
17   Build a synthetic dictionary between $\mathbf{Z}$ and $\mathbf{Y}$ ;
18   Calculate new embedding representation for each word in $\mathbf{Z}$ according to Eq.13;
19   Build a synthetic dictionary between $\mathbf{Z}$ and $\mathbf{Y}$ ;
20   Calculate new embedding representation for each word in $\mathbf{Y}$ according to Eq.13;
21 **end**

---

MUSE dataset (Lample et al., 2018). The Muse dataset consists of monolingual fastText (Bojanowski et al., 2017) embeddings of 300 dimensions trained on Wikipedia monolingual corpus and dictionaries for 110 language pairs. According to (Patra et al., 2019), etymologically close language pairs have lower Gromov Hausdorff (GH) distance compared to etymologically distant languages. Therefore, we evaluate English (En) from/to 4 etymologically close languages with low GH distance: Spanish (Es), French (Fr), Italian (It), and German (De); and 4 etymologically distant languages: Russian (Ru), Chinese (Zh), Hungarian (Hu), and Danish (Da) with high GH distance. Comparing both etymologically close and distant language pairs, we can thoroughly test our model performance with varying degrees of isomorphism between monolingual spaces.

**Baselines** We compare our proposed SA-GAN with several SOTA BLI baselines, including supervised/semi-supervised methods (Lample et al., 2018; Artetxe et al., 2018a; Jawanpuria et al., 2019; Glavaš and Vulić, 2020; Mohiuddin et al., 2020; Ganesan et al., 2021; Tian et al., 2022),

and unsupervised methods (Lample et al., 2018; Artetxe et al., 2018b; Bai et al., 2019; Mohiuddin and Joty, 2019; Ren et al., 2020; Xiong and Tao, 2021). Please refer to Appendix A for the details. For each baseline model, we report the results in the original papers and conduct experiments with the publicly available code if necessary.

**Implementation details** Following previous work, vocabularies of each language are trimmed to the most frequent 200k word embeddings for evaluation, same for the graph generation in section 2.1. The adversarial model uses 75k most frequent words in each language to feed the discriminator. The original word embeddings are normalized following (Artetxe et al., 2018b), including length normalization, center normalization and length normalization again to ensure the word embeddings have a unit length. The generator $G$ is a single linear layer. The discriminator is a multilayer perceptron with two hidden layers of size 2048 and Leaky-ReLU activation functions. We train our models using stochastic gradient descent (SGD), with a batch size of 32, and a learning rate of 0.1. A smoothing coefficient s = 0.1 is added to the discriminator predictions. We train the discriminator more frequently (5 times) than the generator. For the PLM algorithm, the temperature $\tau$ is set to 0.1; the updating rate $p$ is set to 0.02; the vocabulary most frequency $K_m$ is set to 20,000 in the synthetic dictionary; the number of neighbor words as anchors $K_a$ is set to 150; the number of iterations is 10.

### 3.2 Experimental Results

We report the BLI performance over four etymologically close language pairs(en-es, en-fr, en-it, and en-de) and four etymologically distant pairs (en-ru, en-da, en-hu, en-zh) from the MUSE dataset. The results are presented in Table 1. For our approach, we map the embedding representations of the source language (English) into target embedding space (other languages) and evaluate the performance of our model in both directions with the corresponding test datasets. It should be noted that all results reported in the paper are an average of 5 runs. The 'NA' indicates the authors did not report the number or their code is not publicly available, and '*' indicates that the methods fail to converge.

Table 1 shows the Gromov-Hausdorff (GH) distance of the selected language pairs. From the measurements, we can see that etymologically

| | En-Es | | En-Fr | | En-It | | En-De | | En-Ru | | En-Da | | En-Hu | | En-Zh | |
|---|---|---|---|---|---|---|---|---|---|---|---|---|---|---|---|---|
| | → | ← | → | ← | → | ← | → | ← | → | ← | → | ← | → | ← | → | ← |
| GH Distance | 0.31 | | 0.24 | | 0.35 | | 0.29 | | 0.40 | | 0.44 | | 0.40 | | 0.83 | |
| *Sup/Semi-supervised* | | | | | | | | | | | | | | | | |
| (Lample et al., 2018) | 81.4 | 83.2 | 81.1 | 82.4 | 77.3 | 77 | 73.7 | 72.6 | 51.7 | 63.7 | 56.3 | 67.3 | 53.3 | 64.8 | 42.7 | 36.7 |
| (Artetxe et al., 2018a) | 81.9 | 83.4 | 82.1 | 82.4 | 77.4 | 77.9 | 73.5 | 73.5 | 50.5 | 67.3 | 64.1 | 69 | 56.1 | 67.7 | 32.3 | 43.4 |
| (Jawanpuria et al., 2019) | 81.9 | 85.5 | 82.1 | 84.2 | 77.8 | 80.9 | 74.9 | 76.7 | 52.8 | 67.6 | 63.1 | 72.6 | 57 | 69.5 | 49.1 | 45.3 |
| (Glavaš and Vulić, 2020) | 82.4 | 86.3 | 84.5 | 84.9 | 80.2 | 81.9 | 76.5 | 77.5 | 57 | 67.1 | 59.4 | 70 | 55.2 | 70.1 | 47.9 | 47.2 |
| (Mohiuddin et al., 2020) | 82.9 | 86.4 | 82.7 | 84.2 | 78.1 | 81.4 | 75.5 | 75.9 | 52.3 | 67.8 | 60.9 | 70.5 | 57.5 | 66.9 | 42.9 | 42.0 |
| (Ganesan et al., 2021) | 83.1 | 83.3 | 83.7 | 82.9 | 78.6 | 76.1 | 76.1 | 74.7 | 55.8 | 68.7 | NA | NA | NA | NA | NA | NA |
| (Tian et al., 2022) | 84.1 | 86.1 | 83.5 | 84.3 | 79.3 | 81.9 | 76.5 | 72.9 | 58.1 | 68 | 59.3 | 69.1 | 57.5 | 65.9 | 51.7 | 45.9 |
| *Unsupervised* | | | | | | | | | | | | | | | | |
| (Lample et al., 2018) | 81.7 | 83.3 | 82.3 | 81.1 | 77.4 | 76.1 | 74 | 72.2 | 44 | 59.1 | 57.5 | * | 53.5 | 63.7 | 32.5 | 31.4 |
| (Artetxe et al., 2018b) | 82.3 | 84.7 | 82.3 | 83.6 | 78.8 | 79.5 | 74.9 | 74.1 | 49.1 | 65.5 | 64.5 | 67.9 | 56.4 | 67 | 37.5 | 37.8 |
| (Bai et al., 2019) | 82.3 | 84.3 | 82.5 | 83.7 | 78.4 | 77.9 | 74.9 | 73.5 | 49 | 65.8 | 57.7 | 64.6 | 52.5 | 63 | **43.4** | 36.7 |
| (Mohiuddin and Joty, 2019) | 82.7 | 84.7 | 82.8 | 83.7 | 79 | 79.6 | 75.4 | 74.3 | 46.9 | 64.7 | 64.5 | 66.8 | 56.1 | * | * | * |
| (Xiong and Tao, 2021) | 82.8 | 83.9 | 82.5 | 82.3 | 78.6 | 77.9 | 75.3 | 77.9 | 47 | 63.2 | 58.3 | 64.6 | 52.3 | 63.1 | 37.8 | 35 |
| (Ren et al., 2020) | 82.9 | 85.3 | 82.9 | 83.9 | 79.1 | 79.9 | 75.3 | **76.1** | 49.7 | 64.7 | NA | NA | NA | NA | 38.9 | 35.9 |
| (Oprea et al., 2022) | 83.3 | **85.4** | **83.4** | 84.1 | NA | NA | 75.8 | 75.8 | 49.5 | 64.0 | NA | NA | NA | NA | NA | NA |
| SA-GAN without PLM | 83.1 | 85.3 | 82.9 | 84 | 80.2 | 79.9 | 76.9 | 74.7 | 49.7 | 64.9 | 67 | 70.6 | 59.8 | 70.1 | 41.5 | 40 |
| SA-GAN | **84.2** | **85.4** | 83.1 | **84.4** | **81.4** | **80.4** | **77.5** | **76.1** | 50.9 | **66.2** | **69.1** | **71.9** | **60.5** | **71.4** | 41.6 | **42.1** |

Table 1: Word translation accuracy (Precision@1) on MUSE dataset. For each metric, underline marks the highest accuracy among all approaches; **bold** marks the best performance across all unsupervised methods; 'NA' indicates the authors did not report the number or their code is not available; '*' indicates that the methods fail to converge.

close language pairs have lower GH distances compared to etymologically distant ones. We compare SA-GAN with both existing unsupervised and semi/supervised approaches. From Table 1, one can clearly see that our proposed method significantly outperforms previous unsupervised methods over most language pairs, and also obtain comparable performance on the rest. Compared with state-of-the-art unsupervised methods, SA-GAN performs better on 14 of 16 language pairs, especially on en-it and en-de with the absolute improvements of 2% to 2.3%, and on etymologically distant language pairs like en-hu and en-da, with absolute improvements of 4.1% to 4.6% over the best baseline.

Furthermore, compared with the supervised methods, SA-GAN can still achieve competitive results and even outperform existing state-of-the-art supervised methods on some language pairs. The performance of our approach on en-it is 81.4%, compared to 80.2% with the best-supervised method. On en-hu, our SA-GAN obtains 60.5%, which is 3% better than the supervised method. Such performance gains demonstrate the superiority of SA-GAN. From table 1, we also find that leveraging the unsupervised pairwise local mapping (PLM) contributes to bilingual lexicon induction, with a gain of 0.7% on average on etymologically close language pairs and 1.2% on distant language pairs, which is remarkable.

From the results, we note that SA-GAN achieves more improvements in etymologically distant languages, where other unsupervised baselines perform poorly or even fail to converge. This is reasonable as we capture much richer semantics by extracting the structure of embedding space with the GNN module, which helps learn a better mapping function compared with other methods. Moreover, since the distributions of different languages deviate and the isomorphic assumption may not be strictly held (Patra et al., 2019; Søgaard et al., 2018), a global-share mapping is not the optimal solution (Tian et al., 2022). In this situation, an unsupervised PLM algorithm is applied to every word to get personalized mappings, which improves the performance further.

| Methods | En-Fi | | En-He | | En-Ro | | Avg |
|---|---|---|---|---|---|---|---|
| | → | ← | → | ← | → | ← | |
| MUSE | 43.7 | 53.7 | 38 | fail | 58.0 | 66.0 | 43.2 |
| VECMAP | 49.9 | 63.5 | 44.6 | 57.7 | 64.2 | 71.8 | 58.6 |
| Adv-M | 49.8 | 65.5 | 46.1 | 58.6 | 62.6 | 71.9 | 59.1 |
| Adv-O | 49.9 | 65.5 | 46.6 | 59.1 | 65.4 | **74.3** | 60.1 |
| w/o PLM | 52.3 | 64.9 | 47.1 | 57.5 | 66.2 | 72.3 | 60.1 |
| SA-GAN | **52.7** | **66.0** | **48.0** | **59.7** | **66.4** | 72.7 | **60.9** |

Table 2: Word translation accuracy (Precision@1) of morphologically rich languages on MUSE dataset. **Bold** marks the best performance across all methods.

### 3.3 Results of Morphologically Rich Languages

To better explore our model's robustness, we further evaluate our method on "difficult" morphologically rich languages, where unsupervised bilingual dictionary induction performs much worse (Søgaard et al., 2018). Following Oprea et al. (2022),

we evaluate English (En) from/to 3 morphologically rich languages like Finnish(Fi), Hebrew(He), and Romanian(Ro), a mixture of isolating or exclusively concatenating languages from a morphological point of view (Søgaard et al., 2018). Since Lample et al. (2018); Artetxe et al. (2018c); Mohiuddin and Joty (2019); Oprea et al. (2022) consistently perform well, they are selected as baselines for the remaining experiments, denoted as MUSE, VECMAP, Adv-M and Adv-O respectively. The results are shown in Table 2. From the measurements, we can see that our approach outperforms existing methods on 5 of 6 tasks on morphologically rich language pairs, with a gain up to 2.8% on en-fi and 0.8% on average of all languages, which further shows the robustness and effectiveness of our framework.

### 3.4 Ablation Study

To further analyze our approach, we perform ablation studies and measure the contribution of each novel component that is proposed in this work. We conduct extensive ablation on 8 translation tasks from 4 language pairs from the MUSE dataset, consisting of 2 etymologically close and 2 etymologically distant languages.

**Strcture extraction and adversarial training** Here we study the impact of the structure extraction (GNN) module and orthogonality constraint. To avoid the influence of PLM, ablation studies are investigated in the setting without the PLM module, as shown in table 3. One can see that model performance will consistently drop in all language pairs and even fail to converge in distance language pairs if we further remove the GNN module. After enforcing an orthogonality constraint, the performance pairs drop (eg. en-ru) and fail to converge (eg. en-zh) in the distant language pairs that are far from isometry. We can get the following conclusions: 1) the GNN module can capture much richer semantics by extracting structure information of embedding space, which contributes to learning a better mapping function and stable the BLI performance; 2) a strict orthogonality constraint limits the performance of language pairs that are etymologically distant and far from isometry.

**Pair-wise local mapping** Here we aim to study the importance of the designed PLM algorithm, and the influence of updating rate $p$, coefficient scaling, dictionary frequency cutoff, and bidirectional forwarding components to the PLM module.

| Setting | en-it | | en-de | | en-ru | | en-zh | |
| --- | --- | --- | --- | --- | --- | --- | --- | --- |
| | $\rightarrow$ | $\leftarrow$ | $\rightarrow$ | $\leftarrow$ | $\rightarrow$ | $\leftarrow$ | $\rightarrow$ | $\leftarrow$ |
| SA-GAN | 80.2 | **79.9** | **76.9** | **74.7** | **49.7** | **64.9** | **41.5** | **40** |
| w/o GNN | 80.0 | 79.5 | 76.5 | 74.4 | fail | fail | fail | fail |
| constraint | **80.3** | 79.7 | 76.7 | 74.4 | 47.1 | 63.3 | fail | fail |

Table 3: Ablation study on adversarial training.

| Setting | en-it | | en-de | | en-ru | | en-zh | |
| --- | --- | --- | --- | --- | --- | --- | --- | --- |
| | $\rightarrow$ | $\leftarrow$ | $\rightarrow$ | $\leftarrow$ | $\rightarrow$ | $\leftarrow$ | $\rightarrow$ | $\leftarrow$ |
| baseline | 80.2 | 79.9 | 76.9 | 74.7 | 49.7 | 64.9 | 41.5 | 40 |
| SA-GAN | **81.4** | **80.4** | **77.5** | **76.1** | **50.9** | **66.2** | **41.6** | **42.1** |
| w/o Scale | 80.4 | 80.1 | 76.6 | 75.2 | 49.6 | 65 | 40.8 | 40.5 |
| w/o Cutoff | 80.2 | 79.3 | 76.5 | 74.9 | 49.2 | 64.1 | 41.1 | 40.3 |
| w/o Bi | 80.9 | 80.1 | 77.0 | 75.8 | 49.7 | 65.7 | 41.3 | 41.8 |
| w/o $p$ | 68.2 | 63.6 | 65.6 | 56.6 | 31.8 | 39.9 | 28.7 | 32.2 |

Table 4: Ablation study on the pairwise local mapping. Bi: bidirectional forwarding; $p$: updating rate (set to 1 when removed).

The obtained results are presented in Table 4. The baseline in the table is a variant of our approach without using PLM. From the table, we can find that the performance declines over all tasks after removing PLM, revealing the importance of personalized local mappings. As for the different PLM components, we observe that coefficient scaling is necessary to avoid the potential noise introduced by anchor words. The dictionary frequency-based cutoff also has a positive influence on our model, with a 1.2% gain in en-it and 1.3% gain in en-ru. At the same time, the updating rate plays a critical role in the systems. Without updating rate ($p = 1$), the model performance declines sharply due to overly drastic updates of embeddings. Bidirectional forwarding is also beneficial, which provides an optimal solution by mapping the source and target languages together to a latent space, rather than fixing one of them. In summary, every component of PLM is indispensable to achieving better performance.

### 3.5 Parameter Sensitivity Analysis

We further analyze the performance of PLM with respect to two core hyper-parameters: (1) the vocabulary cutoff with the most frequency $K_m$ for synthetic dictionary, and (2) the scaling temperature $\tau$ in Formula 12. The sensitivity analysis is conducted on the en→it language pair on the MUSE dataset.

**Frequency-based vocabulary cutoff** The hyper-parameter $K_m$ denotes the number of most frequent words in each language considered when inducing the synthetic dictionary. As shown in Figure 2(a), On the one hand, when $K_m$ is too small, the syn-

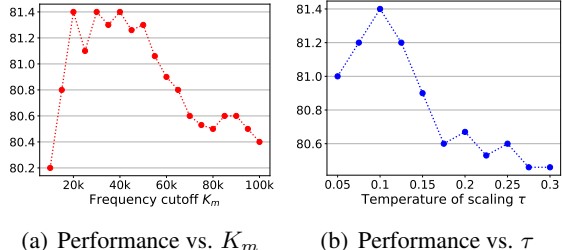

(a) Performance vs. $K_m$      (b) Performance vs. $\tau$

Figure 2: Paramesensitivity analysis.

thetic dictionary can't obtain enough information to guide the local mapping; On the other hand, when $K_m$ is too large, much noise will be introduced, which reduces the quality of the dictionary and declines the performance.

**Temperature of scaling** Figure 2(b) illustrates how the performance varies with different values for scaling temperature $\tau$. We can find that a small $\tau$ helps to increase the influence of the nearest anchors in the dictionary and decrease for the distant ones, which scales the importance coefficients further to provide useful guidelines and reduce the potential noise in the dictionary.

## 4 Conclusion

In this paper, we proposed a novel unsupervised framework SA-GAN for bilingual lexicon induction. Different from previous works that generally treat words in the embedding space as isolated entities, SA-GAN considers each embedding space as a graph and utilizes a GCN module to learn the topological information between words. Additionally, SA-GAN employs a GAN to learn a linear mapping matrix without imposing an orthogonality constraint, thereby transforming both languages into the same embedding space. To further improve the performance, especially for the language pairs where the isomorphic assumption may not hold exactly, we propose a pairwise local mapping algorithm to learn word-specific transformations instead of only applying a shared global mapping to all words. Extensive experiments conducted on the MUSE dataset demonstrate the superior performance of our model. SA-GAN outperforms existing unsupervised alternatives and even surpasses state-of-the-art supervised methods, especially for etymologically distant language pairs.

## Limitations

Although our approach can achieve impressive performance, there are still some limitations to be resolved in the future.

- SA-GAN requires tuning more hyper-parameters compared to previous methods, which is time-consuming.

- SA-GAN matches source and target languages by mapping the source embeddings into the target embedding space, rather than mapping them into a common latent space. While the performance relies on the target word embedding space, the mapping function might be sub-optimal.

- Additionally, SA-GAN focuses on aligning single-word embeddings, making it unsuitable for directly applying to the alignment of multi-word expressions that encompass intricate semantic concepts.

## Acknowledgements

This work was supported by the National Natural Science Foundation of China under Grant 62276101 and the National Key R&D Program of China under Grant 2019YFC1510400, 2022YFC3006405.

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

# A   Related Work

The basic idea for Bilingual lexicon induction (BLI) is to learn cross-lingual mappings which transform word embeddings of different languages to the same embedding space, and then induce Bilingual lexicons from the learned cross-lingual embeddings. Based on the availability of a seed dictionary, we divide related work into the following two categories: supervised/semi-supervised methods and unsupervised methods.

## A.1   Supervised/Semi-supervised Methods

Mikolov et al. (2013) first observe that the word embedding space of one language can be transformed into another using linear mapping, based on the isomorphic assumption that monolingual word embeddings exhibit similar geometric properties across languages. Artetxe et al. (2018a) propose a multi-step framework that generalizes a substantial body of previous work. The core steps include normalization, whitening, orthogonal mapping, reweighting, de-whitening, and dimensionality reduction. Joulin et al. (2018) use a supervised method RCSLS which optimizes the CSLS distance in an end-to-end manner for the supervised matching pairs. Jawanpuria et al. (2019) propose to map both the source and target word embeddings to the common latent space via two orthogonal transformations.

Previously methods learned global-shared linear transformations based on the isomorphic assumption. However, several researchers have found that the isomorphic assumption may not hold all the time, especially for distant language (Søgaard et al., 2018). Patra et al. (2019) observe that the language pairs with high Gromov-Hausdorff (GH) distance cannot be aligned well using orthogonal transformation and proposed semi framework which relaxed isomorphic assumption by jointly optimizing a weak orthogonality constraint in the form of a back-translation loss. Mohiuddin et al. (2020) design a semi-supervised model that uses non-linear mapping in the latent space to learn cross-lingual word embeddings, which is also independent of the

isomorphic assumption. Glavaš and Vulić (2020) propose a supervised word-specific transformation after learning a single global rotation matrix, thus the final mapping function is globally non-linear which performs well in distant language pairs. The PLM algorithm in this paper is inspired by the literature, but differs in that, in comparison with this work, we propose a different transformation framework that can be applied to the unsupervised approach without any labeled data. Sachidananda et al. (2021) align embeddings to isomorphic vector spaces, using pairwise inner products. Li et al. (2022) improve word translation via two-stage contrastive learning. Tian et al. (2022) propose a ranking-based bilingual lexicon induction model which provides sufficient discriminative capacity to rank the candidates.

Nevertheless, all these methods still require supervised signals and cannot be applied to the unsupervised learning setting without any labeled dictionary.

## A.2 Unsupervised Methods

Recently fully unsupervised methods have been proposed to induce a bilingual dictionary by aligning monolingual word embedding spaces. A typical research line is based on adversarial training. Miceli Barone (2016) proposes an adversarial autoencoder framework to map the source language word embeddings to the target language, where an encoder aims to make the transformed embeddings not only indistinguishable by the discriminator but also recoverable after a reversed mapping by the decoder. Although promising, the reported performance is not satisfying. Lample et al. (2018) are the first to show very impressive results for unsupervised word translation where a rough rotation matrix is first learned using the adversarial framework and further refined with a self-learning process. Based on the previous work (Lample et al., 2018), Chen and Cardie (2018) propose an adversarial training framework in the multilingual setting which not only considers one pair of languages at a time but explicitly exploits the relations between all language pairs. Mohiuddin and Joty (2019) revisit the adversarial autoencoder for unsupervised word translation and includes cycle consistency and input reconstruction constraints to guide the mapping. Xiong and Tao (2021) propose an unsupervised approach via bidirectional feature mappings based on cycle-GAN and hybrid training. In contrast to

other frameworks which focus on direct or bidirectional mappings between the source language and target language, Bai et al. (2019) train two autoencoders jointly to transform the source and the target monolingual word embeddings into a shared embedding space to capture the cross-lingual features of word embeddings. Li et al. (2021) observe that low-frequency words tend to be densely clustered in the embedding space, to overcome this issue, they introduced a noise function to disperse dense word embeddings and a Wasserstein critic network to preserve the semantics of the source word embeddings.

On the other hand, non-adversarial approaches have also been proposed for unsupervised crosslingual word alignment. Hoshen and Wolf (2018) use the principal component of monolingual word embeddings to build initial alignment and then iteratively refined the alignment using a variation of the e Iterative Closest Point (ICP) method used in computer vision. Artetxe et al. (2018b) explore the similarity of the embeddings to learn an initial dictionary in an unsupervised way and improve it with a robust self-learning approach. Alvarez-Melis and Jaakkola (2018) cast the problem as an optimal transport problem and measure the similarity between pairs of words across languages using Gromov-Wasserstein distance. Cao and Zhao (2018) propose to use the Coherent Point Drift (CPD) algorithm to map the whole source embeddings to the target embedding space. Inspired by Cao and Zhao (2018), Oprea et al. (2022) employ the CPD algorithm to perform an iterative two-step refinement on the initial global mapping trained by CycleGAN. However, both of them focus on global mapping under the isomorphic assumption. Ren et al. (2020) leverage the Bron-Kerbosch (BK) algorithm to extract clique-level information, which is not only semantically richer than what a single word provides but also reduces the bad effect of the noise in the pre-trained embeddings.

## B Supplementary Experiments

### B.1 Case Study

To better demonstrate the effectiveness of our model on bilingual lexicon induction, we give some examples of the dictionary inferred with our method, comparing with that inferred by two adversarial methods Mohiuddin and Joty, 2019 and Bai et al., 2019, denoted as Adv-M and Adv-B respectively. We choose the language pair English-Danish

as examples, as shown in Table 5.

| Query | Adv-M | Adv-B | Ours | Gold |
|---|---|---|---|---|
| accuracy | nøjagtighed | nøjagtighed | nøjagtighed | nøjagtighed |
| thermal | termiske | varmeledning | termiske | termiske |
| raiders | raiders | panthers | raiders | raiders |
| smoke | flammer | røg | røg | røg |
| kitchen | køkkenhaven | køkkenet | køkkenet | køkkenet |

Table 5: Word translation examples for English-Danish.

In the first example, both approaches find the correct translations. In the following four examples, our approach SA-GAN successfully induces the correct translations with similar meanings, while Adv-M and Adv-B fail to find all correct translations for the given queries, even having significantly different meanings for their induced words compared with the gold translations. From these examples, we find that our method produces bilingual lexicons with higher quality. This is because our approach can effectively utilize the topological structure of the embedding spaces, and pair-wise mapping is learned for every different word to alleviate the reliance on isometry, which improves the BLI performance even further.

## B.2 Downstream Tasks

To better test our model's robustness and effectiveness, we include more downstream tasks, i.e., Semantic Word Similarity and Sentence Translation Retrieval tasks as in the lample2018word and oprea-etal-2022-multi.

**Semantic word similarity** We evaluate the quality of cross-lingual embeddings with the task of Semantic Word Similarity, which aims at evaluating how well the cosine similarity between words of different languages correlates with human-annotated word similarity scores. As shown in Table 6(a), our proposed SA-GAN has a better Pearson's correlation to human-annotated scores across languages on the en-de and de-en language pairs and achieves comparable performance on en-es and es-en, indicating that our model provides good alignment across languages.

**Sentence translation retrieval** This task goes from word to sentence level and studies sentence translation retrieval. Following (Lample et al., 2018), the sentences are represented as a bag of words, and the IDF-weighted average of word embeddings of the sentence is used as its sentence embedding. The closest sentence from the target language is returned as its translation of the given source sentence. Table 6(b) shows sentence translation retrieval results on the Europarl corpus. On the en-fr language pairs, our model obtains the best score with up to 3.5% improvements. Besides, our proposed method performs the best on the averaged accuracy, which depicts that SA-GAN provides better performance in sentence translation retrieval tasks.

| Methods | En-De → | En-De ← | En-Es → | En-Es ← | Avg |
|---|---|---|---|---|---|
| (Lample et al., 2018) | 70.8 | 71.3 | 71.2 | 71.1 | 71.1 |
| (Artetxe et al., 2018b) | 71.9 | 71.9 | 72.1 | 72.1 | 72.0 |
| (Mohiuddin and Joty, 2019) | fail | 72.0 | 72.4 | 71.8 | 72.1 |
| (Oprea et al., 2022) | **72.6** | 72.5 | **73.0** | **72.8** | **72.7** |
| SA-GAN | **72.6** | **72.6** | 72.5 | 72.5 | 72.6 |

(a)

| Methods | En-Fr → | En-Fr ← | En-Es → | En-Es ← | Avg |
|---|---|---|---|---|---|
| (Lample et al., 2018) | 69.1 | 69.9 | 75.1 | 73.9 | 72.0 |
| (Artetxe et al., 2018b) | 69.6 | 69.3 | 74.7 | 74.4 | 72.0 |
| (Mohiuddin and Joty, 2019) | 68.0 | 71.0 | 75.0 | 75.7 | 72.4 |
| (Oprea et al., 2022) | 70.2 | 70.9 | **76.7** | **76.3** | 73.5 |
| SA-GAN | **73.7** | **74.4** | 75.4 | 76.0 | **74.9** |

(b)

Table 6: Performance for (a) Pearson's correlation score(%) for *word similarity* task, and (b) *sentence translation* retrieval accuracy.