# OpenReview forum: "A Structure-Aware Generative Adversarial Network for Bilingual Lexicon Induction"
_EMNLP/2023/Conference — EMNLP 2023 Findings_

### Official Review · Reviewer_ggx5 · 2023-08-03

**Typos Grammar Style And Presentation Improvements:** 1. Line 066
**Soundness:** 4

**Excitement:**

4: Strong: This paper deepens the understanding of some phenomenon or lowers the barriers to an existing research direction.

**Missing References:**

None.

**Paper Topic And Main Contributions:**

This paper proposes a GAN based approach to align word embedding spaces for the Bilingual Lexical Induction (BLI) task. The paper proposes a graphical convolutional network (GCN) to learn the topology of the embedding space in a given language and then utilizes a GAN to learn cross space correlations. They showcase the performance of the method of a diverse set of datasets.

*Main Contributions*
1. The paper proposes a new approach for Bilingual Induction between language pairs. They use a GCN to learn intra-space topology and a GAN to map two spaces together.
2.  Additionally, they propose a pairwise local mapping algorithm to train learn embeddings based on their nearest neighbors.
3. The paper has experiments on 16 languages pairs.

**Questions For The Authors:**

1. Line 213: Between noes I and j, the negative log of the euclidean distance is computed. Can the authors explain the reasoning here?
2. Line 371-373: Why are words with low frequency filtered out?
3. Line 391: If anchor points that are far off have a high coefficient of 0.4, will the softmax output (when scaled) be high too? If baseline values (points which are far away) are high, then the probability will closer to a uniform distribution.
4. In Table 2: Why does the GAN model fail if there isn’t a GNN?
5. In Table 3: What does “without p” mean? Is it set to 1?

**Reasons To Accept:**

1. The paper is written well with few mistakes. Some corrections have been suggested.
2. The authors propose a method that exploits the manifold structure in the embedding space in order to learning a mapping between two embedding spaces.
3. The papers have extensive experiments that analyze different aspects of the pipeline and show the importance of each individual component.

**Reasons To Reject:**

1. From algorithm 1, we can see that model training is an amalgamation of several strategies in comparison to a singular outlined training algorithm.
2. Although isomorphism is not an underlying assumption, the learned transformation matrix is linear. In order to avoid the pitfalls, a non-linear mapping can be considered.
3. Although the paper shows improvements, it does not ground it. For example, what does it mean to have a 2% relative improvement on en-it in comparison to the state of the art method. The paper can benefit from a small section on error analysis or comparison with an existing SOTA model to outline what the improvement is.

**Reproducibility:**

3: Could reproduce the results with some difficulty. The settings of parameters are underspecified or subjectively determined; the training/evaluation data are not widely available.

**Reviewer Confidence:**

4: Quite sure. I tried to check the important points carefully. It's unlikely, though conceivable, that I missed something that should affect my ratings.

---

> ### Author Rebuttal · Authors · 2023-08-27
>
> # Response to *Reviewer ggx5*
>
> ## Comment1 :
> From algorithm 1, we can see that model training is an amalgamation of several strategies in comparison to a singular outlined training algorithm.
> ### Response:
> Thank you for reviewing and providing valuable comments. We adopt this pattern for several reasons:
>
> 1, Each module has its own role to play while targeting different goals. By the strategy of splitting, each module can focus more on its task and improve the overall performance while reducing the complexity. Specially, the GCN module aims to extract structural and semantical information of the embeddings and provide prior knowledge for subsequent GANs to keep the training stable especially for languages in low-isomorphism; the GANs module aims to learn a global mapping, which roughly aligns the word embeddings of two languages, only then can the PLM further fine-turn the embeddings in local area according to the seed dictionary induced by the aligned embeddings; the PLM module aims to learn word-specific local transformations instead of only applying a shared global mapping to all words. Therefore, the final mapping relation between the two languages is non-linear in global, which avoids the pitfalls of isomorphism. We can observe from Table 1, each three module focuses on its task and performs better on 15 of 16 language pairs, especially on etymologically distant language pairs like en-hu and en-da, with improvements of 4.1% to 4.6% in absolute over the best unsupervised baseline, which proves the effectiveness of our strategies.
>
> 2, The amalgamation of several strategies is very widely adopted in Bilingual Lexicon Induction (BLI) models. Artetxe et.al. [1] propose a multi-step framework which generalizes previous work. Their framework consists of several steps: Normalization, whitening, mapping, re-weighting, de-whitening, and dimensionality reduction. Other works can be considered as a combination of different steps. For most unsupervised methods shown in Table 1, their methods consist of normalization, mapping, re-weighting and de-whitening. i.e., Artetxe et.al [2], Bai et.al. [3] Mohiuddin [4]. Moreover, Oprea et.al. [5] propose an unsupervised multi-stage framework for learning bi-lingual word alignment, by using a combination of adversarial training, refinement procedure, and point set registration, which is also an amalgamation of several strategies.
>
> 3, Through modularization, our method can be more easily split and used by other researchers as part of the expansion of their models. For example, our unsupervised PLM module can be adopted to any other existing unsupervised methods as it does not rely on any supervised signal. They only need to feed their output embeddings to the PLM module to get the upgraded embeddings and improve their performances on low-isomorphism languages.
>
>
> [1] Artetxe, Mikel, Gorka Labaka, and Eneko Agirre. "Generalizing and improving bilingual word embedding mappings with a multi-step framework of linear transformations." Proceedings of the AAAI Conference on Artificial Intelligence. Vol. 32. No. 1. 2018.
>
> [2] Artetxe, Mikel, Gorka Labaka, and Eneko Agirre. "A robust self-learning method for fully unsupervised cross-lingual mappings of word embeddings." Proceedings of the 56th Annual Meeting of the Association for Computational Linguistics (Volume 1: Long Papers). 2018.
>
> [3] Bai, Xuefeng, et al. "A bilingual adversarial autoencoder for unsupervised bilingual lexicon induction." IEEE/ACM Transactions on Audio, Speech, and Language Processing 27.10 (2019): 1639-1648.
>
> [4] Mohiuddin, Muhammad Tasnim, and Shafiq Joty. "Revisiting Adversarial Autoencoder for Unsupervised Word Translation with Cycle Consistency and Improved Training." Proceedings of the 2019 Conference of the North American Chapter of the Association for Computational Linguistics: Human Language Technologies, Volume 1 (Long and Short Papers). 2019.
>
> [5] Oprea, Silviu Vlad, Sourav Dutta, and Haytham Assem. "Multi-Stage Framework with Refinement Based Point Set Registration for Unsupervised Bi-Lingual Word Alignment." Proceedings of the 29th International Conference on Computational Linguistics. 2022.
>
>
> ## Comment2 :
> Although isomorphism is not an underlying assumption, the learned transformation matrix is linear. In order to avoid the pitfalls, a non-linear mapping can be considered.
> ### Response:
> We thank the reviewer for the careful review and insightful comments. Previous methods learn an  **orthogonal** linear mapping under the isomorphism assumption. As the assumption does not hold, we remove the orthogonality constraint of the mapping function and an improvement is achieved, as shown in Table 2 in original paper.
>
> However, after replacing the transformation with a non-linear one (MLP with tanh as the non-linear activation function), our model’s performance consistently drops in all language pairs and the stability of the model is severely degraded in low-isomorphism languages, as shown in Table A below.
>
> To exclude the influence of PLM, the results are investigated in the setting without the PLM module. We run 5 times for each language and only report the average of success runs. “P@1” indicates the BLI accuracy and “S” is the number of successful runs with >5% accuracy in 5 runs. The results show that, the performance is slightly but consistently drop in etymologically close languages (i.e., en-es and en-it). While for the low-isomorphism language pairs (i.e., en-zh and en-ru), not only the performance drops, but also the stability is severely degraded.  The original SAGAN can successfully run 5 times in 5 runs. However, after replacing the transformation with a non-linear one, it can only successfully run 1 or 2 times in 5 runs and the performance also drops. Therefore, we use the linear mapping in the paper.
>
> |Type|en-es| |en-it	||	en-zh	||	en-ru	||
> |---|---|---|---|---|---|---|---|---|
> ||P@1	|S|	P@1|	S|	P@1|	S	|P@1|	S|
> |Linear	|	83.1|	5	|80.2|	5|	41.5|	5|	49.7|	5|
> |Nonlinear		|82.9	|5	|80.1|	5	|39.5|	1	|49.5	|2|
>
> We think there are several reasons under this phenomenon:
>
> 1, After adding the activation function to the mapping, the geometric structure of the generated embeddings is changed too much, and the distribution is quite different between the mapped source embeddings and the target language. Therefore, it is difficult for the generator to generate enough real samples to fool the discriminator, and the adversarial training cannot continue.
>
> 2, It is more suitable for supervised methods to use a non-linear mapping. Indeed, some methods [1-2] have found that the model can be trained successfully and even performs better on low isomorphism language pairs after adding non-linear mapping. However, all of these methods are supervised methods, where supervised signals are indispensable. As for unsupervised methods, they propose many other improvements for the low-isomorphism problem, but still retain the linear mapping, e.g., bai et.al[3] and Mohiuddin[4], which is consistent with the previously common unsupervised method.
>
> Although our transformation matrix is still linear, our full model is globally non-linear. This is because: after the linear transformation matrix is learned and the embeddings of two languages are mapped to the same space (till now, model is linear), we further recompute and update the embeddings for each point according to the local structure during PLM procedure. In other words, through PLM, we learn word-specific local transformations instead of only applying a shared global mapping to all words. Therefore, the final mapping relation between the two languages is non-linear in global, which avoids the pitfalls of isomorphism. We can also observe from Table 1 and the ablation study in Table 3, the performances are improved over all tasks after adding PLM.
>
> [1] Mohiuddin, Muhammad Tasnim, M. Saiful Bari, and Shafiq Joty. "LNMap: Departures from Isomorphic Assumption in Bilingual Lexicon Induction Through Non-Linear Mapping in Latent Space." Proceedings of the 2020 Conference on Empirical Methods in Natural Language Processing (EMNLP). 2020.
>
> [2] Tian, Zhoujin, et al. "RAPO: An Adaptive Ranking Paradigm for Bilingual Lexicon Induction." Proceedings of the 2022 Conference on Empirical Methods in Natural Language Processing. 2022.
>
> [3] Bai, Xuefeng, et al. "A bilingual adversarial autoencoder for unsupervised bilingual lexicon induction." IEEE/ACM Transactions on Audio, Speech, and Language Processing 27.10 (2019): 1639-1648.
>
> [4] Mohiuddin, Muhammad Tasnim, and Shafiq Joty. "Revisiting Adversarial Autoencoder for Unsupervised Word Translation with Cycle Consistency and Improved Training." Proceedings of the 2019 Conference of the North American Chapter of the Association for Computational Linguistics: Human Language Technologies, Volume 1 (Long and Short Papers). 2019.
>
>
> ## Comment3 :
> Although the paper shows improvements, it does not ground it. For example, what does it mean to have a 2% relative improvement on en-it in comparison to the state of the art method. The paper can benefit from a small section on error analysis or comparison with an existing SOTA model to outline what the improvement is.
> ### Response:
> We thank the reviewer for the careful review and valuable suggestions. In our original manuscript, as shown in Section Appendix. B.1, we have made a Case Study of quality of the seed lexicon for En-Da on MUSE dataset. We give some examples of the dictionary inferred with our method and compare with two adversarial methods.From the examples, we can find that our approach SAGAN successfully induces the correct translations with similar meanings, while the other methods fail to find all correct translations for the given queries, even having significantly different meanings for their induced words compared with the gold translations. From these examples, we find that our method produces bilingual lexicons with higher quality.
>
> To better outline what the improvement our model can provide, we further added some experiments on downstream tasks, i.e., **Semantic Word Similarity** and **Sentence Translation Retrieval** tasks as in the Lample et.al[1] and Oprea et.al[2] , which lends more credibility to the robustness and effectiveness of our approach, as shown in table A and table B. For each result, we performed the experiments for 5 times and reported the average results.
>
> |TableA: Sord Sim|En-De|	De-En|En-Es|Es-En|
> |---|---|---|---|---|
> |(Lample et al., 2018b) 	|	0.708|	0.713	|	0.712|	0.711|
> |(Artetxe et al., 2018b) 	|	0.719|	0.719|		0.721|	0.721|
> |(Mohiuddin and Joty, 2019) 	|	fail	|0.72|		0.724|	0.718|
> |SA-GAN without PLM	|	0.721|	0.721	|	0.723|	0.723|
> |SA-GAN	|**0.726**	|**0.726**		|**0.725**	|**0.725**|
>
> For Semantic Word Similarity task, it evaluates the quality of cross-lingual word alignment based on the correlation between cosine similarity between words in different languages and human-annotated word similarity scores. Table A shows that SAGAN achieves a better Pearson’s correlation to human-annotated scores across languages (except it) – providing better alignment across languages.
>
>
> |TableB: Sentence Trans|		en-es|	es-en	|	en-fr|	fr-en|
> |---|---|---|---|---|
> |(Lample et al., 2018b)	|	75.1|	73.9|		69.1|	69.9|
> |(Artetxe et al., 2018b)		|74.7	|74.4		|69.6	|69.3|
> |(Mohiuddin and Joty, 2019)		|75.0|	75.7|		68.0	|71.0|
> |SA-GAN without PLM|		73.3|	73.7	|	71.2|	71.8|
> |SA-GAN 	|	**75.4**|	**76.0**	|	**73.7**|	**74.4**|
>
> For Sentence Translation Retrieval task, it studies sentence translation retrieval on Europarl corpus. The sentences are represented as a bag-of-words and the idf-weighted average of word embeddings is considered as its encoding. For each source sentence, the closest sentence from the target language is returned as its translation. Table B shows sentence translation retrieval result on Europarl corpus, which depicts that SAGAN provides better sentence translation retrieval accuracy, with up to 3.4% score improvements.
>
> From these tables, we see that the performance of Semantic Word Similarity and Sentence Translation Retrieval is restricted to the quality of BLI results. As our method provides better word translations, the downstream tasks benefit from ours accordingly.
>
> [1] Lample G, Conneau A, Ranzato M A, et al. Word translation without parallel data[C]//International Conference on Learning Representations. 2018.
>
> [2] Oprea, Silviu Vlad, Sourav Dutta, and Haytham Assem. "Multi-Stage Framework with Refinement Based Point Set Registration for Unsupervised Bi-Lingual Word Alignment." Proceedings of the 29th International Conference on Computational Linguistics. 2022.
>
>
> ## Comment4 :
> Line 213: Between noes I and j, the negative log of the euclidean distance is computed. Can the authors explain the reasoning here?
> ### Response:
> We thank the reviewers for the comments. By doing this, neighbor nodes closer to node i will have higher weights in adjacency matrix.
>
> In comparison to use “negative Euclidean distance” directly, “negative log“ further scales the distance weights. As in high dimensional space, the nodes are very close to each other. Even node j is “very far” in semantic, it is still close in Euclidean distance (i.e., 0.8), compared to a real semantic close node k (i.e., 0.3 in Euclidean distance). However, after scaling, these two weights are now 0.01 for distant node j and 0.52 for close node k. Therefore, it makes nodes more distinguishable and the local structural information is preserved better by “filtering” these distant neighbors.
>
>
> ## Comment5 :
> Line 371-373: Why are words with low frequency filtered out?
> ### Response:
> In word embedding training (using fasttext or word2vec), those low-frequency words are usually of low quality. In extreme cases, words that occur only once may appear anywhere in the embedding space, containing no semantic information but noise which has a negative effect on the quality of seed dictionary. Therefore, filtering the words with low frequency is performed.
>
>
> ## Comment6 :
> Line 391: If anchor points that are far off have a high coefficient of 0.4, will the softmax output (when scaled) be high too? If baseline values (points which are far away) are high, then the probability will closer to a uniform distribution.
> ### Response:
> We thank the reviewers for the comments. After the softmax output, the coefficient is not high, since we further scale the weight through temperature controlling. For example, there are two anchor nodes i and j with coefficient of 0.8 and 0.4 separately (with 2 times larger). After softmax with temperature=0.1, the new coefficients will be 0.98 and 0.02 (with 49 times larger).
>
> If baseline values are high, in this case, we think these points are the exactly anchors that we want. As the words that are semantically similar tend to be close in embedding space, we think these anchors with high value have similar semantic with the center word, and will have the same margin (distant of the word pair<$x_d$ to $y_d$> in seed dictionary) to move.
>
>
> ## Comment7 :
> In Table 2: Why does the GAN model fail if there isn’t a GNN?
> ### Response:
> For unsupervised models, it is difficult to train and have a good performance on low isomorphism languages (eg, En-Zh and En-Ru). In these languages, the unsupervised models even fail to converge. Same result can be found in Mohiuddin et.al [1-2], which directly use the original word embedding to train the GAN-based model.
>
> Our GNN, on the other hand, can effectively extract structural and semantical information of the embeddings and provide prior knowledge for subsequent GANs, which helps the generator generates better samples and stabilizes the training.
>
> [1] Mohiuddin, Muhammad Tasnim, M. Saiful Bari, and Shafiq Joty. "LNMap: Departures from Isomorphic Assumption in Bilingual Lexicon Induction Through Non-Linear Mapping in Latent Space." Proceedings of the 2020 Conference on Empirical Methods in Natural Language Processing (EMNLP). 2020.
>
> [2] Mohiuddin, Muhammad Tasnim, and Shafiq Joty. "Revisiting Adversarial Autoencoder for Unsupervised Word Translation with Cycle Consistency and Improved Training." Proceedings of the 2019 Conference of the North American Chapter of the Association for Computational Linguistics: Human Language Technologies, Volume 1 (Long and Short Papers). 2019.
>
>
> ## Comment8 :
> In Table 3: What does “without p” mean? Is it set to 1?
> ### Response:
> We greatly appreciate the reviewer for the comments. In our original manuscript, as shown in Line 590, we have denoted the meaning of “without p” and the p is set to 1. For more clarity, the table has been revised as directed in a clearer way and a relevant description is added to the table caption.
>
>
> ## Comment9 :
> induction -> inductive
> ### Response:
> Thank you for pointing out this problem. The word format has been revised as directed.
>
>
> ## Comment10 :
> Is PLM a mapping function or a loss? The authors can clarify if the PLM is a mapping function or an optimization criteria to learn a mapping. The text in the paper and equation appear to be different.
> ### Response:
> Thank you for your advice. PLM is a mapping function which recomputes and upgrades the embedding representations directly through the equation. As suggested, we reclarify the description in Line 345 in the revised version.
>
> ## Comment11 :
> The interpretation of “_” in Line 494 and Table 1 caption is different. The authors can clarify which one to use.
> ### Response:
> We thank the reviewer for the careful review and valuable suggestions. As suggested, we have changed the mark in Line 494 to “NA” to indicate “the authors did not report the number or their code is not publicly available”.
>
> ## Comment12 :
> The authors can add information if the improvements are absolute or relative.
> ### Response:
> Thank the reviewer for advice. The improvements are absolute and the description in Line511 has be revised as suggested.
>
> ## Comment13 :
> “hold” -> “held”.
> ### Response:
> Thank you for pointing out this problem. The Verb tense has been revised as directed. In addition, we have carefully proofread the entire paper to improve the expression quality.

---

### Official Review · Reviewer_vh2V · 2023-08-04

**Soundness:** 4

**Excitement:**

3: Ambivalent: It has merits (e.g., it reports state-of-the-art results, the idea is nice), but there are key weaknesses (e.g., it describes incremental work), and it can significantly benefit from another round of revision. However, I won't object to accepting it if my co-reviewers champion it.

**Missing References:**

1. Improving Word Translation via Two-Stage Contrastive Learning (ACL 2022)
2. Filtered Inner Product Projection for Crosslingual Embedding Alignment (ICLR 2021)

**Paper Topic And Main Contributions:**

This paper focuses on bilingual lexicon induction, presenting a structure-aware approach to effectively model topological structure mapping. The proposed method utilizes a Graph Neural Network (GNN) to generate structure-aware word embeddings and employs a Generative Adversarial Network (GAN) to align two sets of word embeddings. Experimental results demonstrate that the proposed method outperforms several baselines, yielding superior outcomes in the task of bilingual lexicon induction.

**Reasons To Accept:**

1.This paper is well-organized and presented in an easy-to-follow manner.
2.The motivation behind this approach is reasonable and well supported by experiments.
3.The paper demonstrates substantial improvements over several baseline methods.

**Reasons To Reject:**

The experiments is somewhat weak.
1) The main contribution of this paper lies in the structure-aware encoder-based model for seed lexicon induction, there should be an experiment to study the quality of seed lexicon.
2) While the paper focuses on bilingual lexicon induction, it would be beneficial to include a downstream task, such as cross-lingual Natural Language Inference (NLI), to demonstrate the potential impact of the proposed method on downstream applications. This would provide further insights into the effectiveness of the approach beyond the specific lexicon induction task.

**Reproducibility:**

4: Could mostly reproduce the results, but there may be some variation because of sample variance or minor variations in their interpretation of the protocol or method.

**Reviewer Confidence:**

4: Quite sure. I tried to check the important points carefully. It's unlikely, though conceivable, that I missed something that should affect my ratings.

---

> ### Author Rebuttal · Authors · 2023-08-27
>
> # Response to *Reviewer vh2V*
>
> ## Comment1 :
> The main contribution of this paper lies in the structure-aware encoder-based model for seed lexicon induction, there should be an experiment to study the quality of seed lexicon.
> ### Response:
> In our original manuscript, as shown in Section Appendix. B.1, we made a Case Study of quality of the seed lexicon for En-Da on MUSE dataset. We also gave some examples of the dictionary inferred with our method and compare with two adversarial methods. From the examples, we can find that our approach SAGAN successfully induces the correct translations with similar meanings, while the other methods fail to find all correct translations for the given queries, even having significantly different meanings for their induced words compared with the gold translations. From these examples, we find that our method produces bilingual lexicons with higher quality.
>
> We thank the reviewer for the careful review and valuable suggestions. To better analyze the quality of seed lexicon, we will further give some visualization of the retrieval results of our method in a space matter in the revised manuscript.
>
> ## Comment2 :
> While the paper focuses on bilingual lexicon induction, it would be beneficial to include a downstream task, such as cross-lingual Natural Language Inference (NLI), to demonstrate the potential impact of the proposed method on downstream applications. This would provide further insights into the effectiveness of the approach beyond the specific lexicon induction task.
> ### Response:
> We thank the reviewer for the insightful suggestions. According to the suggestion of the reviewer, we have added some experiments by conducting more downstream tasks, i.e., **Semantic Word Similarity** and **Sentence Translation Retrieval** tasks as in the Lample et.al [1] and Oprea et.al [2] , which lends more credibility to the robustness and effectiveness of our approach, as shown in table A and table B. For each result, we performed the experiments for 5 times and reported the average results.
>
> |TableA: Sord Sim|En-De|	De-En|En-Es|Es-En|
> |---|---|---|---|---|
> |(Lample et al., 2018b) 	|	0.708|	0.713	|	0.712|	0.711|
> |(Artetxe et al., 2018b) 	|	0.719|	0.719|		0.721|	0.721|
> |(Mohiuddin and Joty, 2019) 	|	fail	|0.72|		0.724|	0.718|
> |SA-GAN without PLM	|	0.721|	0.721	|	0.723|	0.723|
> |SA-GAN	|**0.726**	|**0.726**		|**0.725**	|**0.725**|
>
> For Semantic Word Similarity task, it evaluates the quality of cross-lingual word alignment based on the correlation between cosine similarity between words in different languages and human-annotated word similarity scores. Table A shows that SAGAN achieves a better Pearson’s correlation to human-annotated scores across languages – providing better alignment across languages.
>
> |TableB: Sentence Trans|		en-es|	es-en	|	en-fr|	fr-en|
> |---|---|---|---|---|
> |(Lample et al., 2018b)	|	75.1|	73.9|		69.1|	69.9|
> |(Artetxe et al., 2018b)		|74.7	|74.4		|69.6	|69.3|
> |(Mohiuddin and Joty, 2019)		|75.0|	75.7|		68.0	|71.0|
> |SA-GAN without PLM|		73.3|	73.7	|	71.2|	71.8|
> |SA-GAN 	|	**75.4**|	**76.0**	|	**73.7**|	**74.4**|
>
> For Sentence Translation Retrieval task, it studies sentence translation retrieval on Europarl corpus. The sentences are represented as a bag-of-words and the idf-weighted average of word embeddings is considered as its encoding. For each source sentence, the closest sentence from the target language is returned as its translation. Table B shows sentence translation retrieval result on Europarl corpus, which depicts that SAGAN provides better sentence translation retrieval accuracy, with up to 3.4% score improvements.
>
> [1] Lample G, Conneau A, Ranzato M A, et al. Word translation without parallel data[C]//International Conference on Learning Representations. 2018.
>
> [2] Oprea, Silviu Vlad, Sourav Dutta, and Haytham Assem. "Multi-Stage Framework with Refinement Based Point Set Registration for Unsupervised Bi-Lingual Word Alignment." Proceedings of the 29th International Conference on Computational Linguistics. 2022.
>
> ## Comment3 :
> Missing References:
>
> Improving Word Translation via Two-Stage Contrastive Learning (ACL 2022)
>
> Filtered Inner Product Projection for Crosslingual Embedding Alignment (ICLR 2021)
>
> ### Response:
> It is a very good suggestion. These references are very helpful for us to understand BLI task and further research on CLE evaluation and model analysis. According to the suggestion of the reviewer, we have added these references in the revised manuscript in Section of Related work.

---

### Official Review · Reviewer_qVkS · 2023-08-07

**Soundness:** 3

**Excitement:**

3: Ambivalent: It has merits (e.g., it reports state-of-the-art results, the idea is nice), but there are key weaknesses (e.g., it describes incremental work), and it can significantly benefit from another round of revision. However, I won't object to accepting it if my co-reviewers champion it.

**Missing References:**

"Multi-Stage Framework with Refinement based Point Set Registration for Unsupervised Bi-Lingual Word Alignment". Oprea et al., COLING 2022.
"Point Set Registration for Unsupervised Bilingual Lexicon Induction". H. Cao and T. Zhao, IJCAI 2018.

**Paper Topic And Main Contributions:**

The paper proposes a framework for bilingual word mapping across the independent language embedding space. The author(s) combine structural information of the embedding spaces via GCN, and then employ GAN to learn the mapping function. Experimental evaluation of the presented framework has been conducted on bilingual dictionary creation benchmark dataset.

**Questions For The Authors:**

My major question is about the GCN - what does it do, if it does not learn anything? How is, then, the structural information propagated within the graph?

**Reasons To Accept:**

1. The paper tackles an important problem in the domain of multilingual NLP, wherein such dictionary induction is vital in certain scenarios.
2. The intuition of directly incorporating embedding space structural information is interesting, and might spark further advancement in this direction.
3. Experimental results depicts the effectiveness of the proposed approach.

**Reasons To Reject:**

1. Although the structural information has not been explicitly used in the current problem statement, it has been implicitly used in few previous works on bilingual mapping induction. Please see:
"Multi-Stage Framework with Refinement based Point Set Registration for Unsupervised Bi-Lingual Word Alignment". Oprea et al., COLING 2022.
"Point Set Registration for Unsupervised Bilingual Lexicon Induction". H. Cao and T. Zhao, IJCAI 2018.
As such, a proper discussion and comparison is necessary in the current paper.

2. The GCN that is proposed has no learnable parameters, so it is a static matrix transformation operation. So, why is the term GCN used - isn't it misleading? Further, if it is not learning anything, it is a simple aggregation function, as far as I understood. Then the structural information is not propagating through the entire graph - this is counter-intuitive because the author(s) claim that structural information usage is a key feature of the proposed framework. Am I missing something here?

3. For experiments, I have 2 comments - (i) addition of performance on word similarity and sentence translation tasks as in the MUSE paper (and others) would lend more credibility to the robustness and effectiveness of the framework. (ii) addition of morphologically rich languages like Finnish, Hebrew, etc and low-resource languages in the experiments would be good to have (minor point).

**Reproducibility:**

3: Could reproduce the results with some difficulty. The settings of parameters are underspecified or subjectively determined; the training/evaluation data are not widely available.

**Reviewer Confidence:**

4: Quite sure. I tried to check the important points carefully. It's unlikely, though conceivable, that I missed something that should affect my ratings.

---

> ### Author Rebuttal · Authors · 2023-08-27
>
> # Response to *Reviewer qVkS*
>
> ## Comment1 :
> "Although the structural information has not been explicitly used in the current problem statement, it has been implicitly used in few previous works on bilingual mapping induction. Please see: "Multi-Stage Framework with Refinement based Point Set Registration for Unsupervised Bi-Lingual Word Alignment". Oprea et al., COLING 2022. "Point Set Registration for Unsupervised Bilingual Lexicon Induction". H. Cao and T. Zhao, IJCAI 2018. As such, a proper discussion and comparison is necessary in the current paper."
> ### Response:
> Thank the reviewer for helpful comment. We have cited the two references and added some discussions as follows.
>
> Both Cao et.al [1] and Oprea et.al [2] propose to use the Coherent Point Drift (CPD) algorithm [3] to preserve the underlying global topological structure by aligning source and target embeddings into a shared embedding space. Specifically, Cao et.al [1] used the CPD algorithm to map the whole source embeddings to the target embedding space. Different from [1], Oprea et.al [2] used a CycleGAN to map them into an initial globally aligned embedding space, and then employed the CPD algorithm to perform an iterative two-step refinement on the initial global mapping. However, they focus on the global mapping under the isomorphic assumption.
>
> Different from their methods:
>
> Firstly, our PLM method captures the local inter-space structural information based on the nearest word pairs neighbors, and learns word-specific transformations instead of a global-shared mapping. Therefore, our method is more suitable for the language pairs of low isometry.	 We can directly see the improvement brought by PLM from the experiments, as shown in table 1. The BLI performance is further improved with PLM, with a gain of 0.7% on average on etymologically close language pairs and 1.2% on low-isomorphic language pairs, which is remarkable.
>
> Secondly, our method explicitly explores intra-space structural correlations using GCN before we train the transformation by GAN (Oprea et al. use CPD after GAN is trained), which provides much rich semantic information to the embeddings and that can be used as a-priori information to the following GAN training. With the help of explicitly intra-space structural information, the generator can generate much more accurate and semantically consistent cross-language word embedding samples, which greatly stabilizes the training process, especially for the missions on low-isomorphic language pairs, where the other GAN-based unsupervised methods (Lample et al[4]. and Mohiuddin et al.[5]) usually fail to converge (eg, En-Fi or En-Zh according to LNMAP[6]). For our method, we can also observe in Table 2 that the performance will consistently drop in all language pairs and even fail to converge in low-isomorphic (distance) language pairs if we further remove the GCN module, which indicates the intra-space structural extraction before adversarial training will greatly contribute to GAN’s training and stabilize the BLI performance.
> | Method | En-Es  | Es-En   | En-De   | De-En   | En-Fr  | Fr-en   | En-Ru    | Ru-En    | Avg   |
> |---|---|---|---|---|---|---|---|---|---|
> |(Oprea et al., 2022)|	83.3|	**85.4**|	75.8|	75.8|	**83.4**|	84.1|	49.5|	64.0|	75.2|
> |SA-GAN|	**84.2**|	**85.4**|	**77.5**|	**76.1**|	83.1|**84.4**	|**50.9**|	**66.2**|	**76.0**|
>
> In addition, as suggested by the reviewer, we further added a comparison with Oprea et.al [1] as suggested in Table 1 of revised manuscript. It can be seen that our approach outperforms the Oprea et.al [2]’s methods on most language pairs (7 of 8 language pairs), with a gain of 0.8% on average and a larger improvement is achieved in low-isomorphic languages (1.4% on En-Ru and 2.2% on Ru-En). The added experiments can prove the effectiveness of our approach. The updated results will be shown in Table 1 in the revision.
>
> We thank the reviewer for providing the relevant references. Making proper discussion and comparison with these references is very helpful for us to understand how the structural information extraction contributes to BLI task. We have added these references in the revised manuscript and more comparison has been made. We also hope our work can spark further advancement in structural information extraction and contributes to widely used languages in world of low isometry.
>
> [1] Cao, Hailong, and Tiejun Zhao. "Point Set Registration for Unsupervised Bilingual Lexicon Induction." IJCAI. 2018.
>
> [2] Oprea, Silviu Vlad, Sourav Dutta, and Haytham Assem. "Multi-Stage Framework with Refinement Based Point Set Registration for Unsupervised Bi-Lingual Word Alignment." Proceedings of the 29th International Conference on Computational Linguistics. 2022.
>
> [3] Myronenko, Andriy, and Xubo Song. "Point Set Registration: Coherent Point Drift." IEEE TRANSACTIONS ON PATTERN ANALYSIS AND MACHINE INTELLIGENCE 32.12 (2010).
>
> [4] Lample, Guillaume, et al. "Word translation without parallel data." International Conference on Learning Representations. 2018.
>
> [5] Mohiuddin, Muhammad Tasnim, and Shafiq Joty. "Revisiting Adversarial Autoencoder for Unsupervised Word Translation with Cycle Consistency and Improved Training." Proceedings of the 2019 Conference of the North American Chapter of the Association for Computational Linguistics: Human Language Technologies, Volume 1 (Long and Short Papers). 2019.
>
> [6] Mohiuddin, Muhammad Tasnim, M. Saiful Bari, and Shafiq Joty. "LNMap: Departures from Isomorphic Assumption in Bilingual Lexicon Induction Through Non-Linear Mapping in Latent Space." Proceedings of the 2020 Conference on Empirical Methods in Natural Language Processing (EMNLP). 2020.
>
>
> ## Comment2 :
> The GCN that is proposed has no learnable parameters, so it is a static matrix transformation operation. So, why is the term GCN used - isn't it misleading? Further, if it is not learning anything, it is a simple aggregation function, as far as I understood. Then the structural information is not propagating through the entire graph - this is counter-intuitive because the author(s) claim that structural information usage is a key feature of the proposed framework. Am I missing something here?
>
> ### Response:
> We thank the reviewer for the careful review and valuable suggestions. For traditional GCN, each graph convolutional layer can be divided into two operations: embedding propagation (EP) and embedding transformation (ET). Concretely, GNN first executes the EP with the normalized adjacency matrix $\hat{A}$ to the embedding matrix $X$:
> $$EP(X)=\hat{A}X,$$
> Then, the propagated feature $\hat{X} = EP(X)$ will be transformed with the learnable transformation matrix $W$ and the activation function $\sigma(·)$:
> $$ET(\hat{X})=σ(\hat{X}W).$$
> For our simplified GCN module, it can be seen as only EP is kept, and the transformation ($W$) and the activation function in ET operation are removed. Therefore, **the structural information can still propagate through the graph through multiple EP operations.**
>
> We propose to use GCN without $W$ for several reasons:
>
> 1, **Removing $W$ in GCN doesn’t mean that it learns nothing.** The core step of GCN is the propagation operation. According to Thomas [1], even the GCN model with random weights (eg, $W=I$) can still serve as a powerful feature extractor for nodes in a graph.
>
> 2, **Recent work shows that the transformation operation ($W$) is not necessary in GCN.** Previous works [2-3] have shown that the true effectiveness of GNNs lies in the EP operation rather than the ET operation inside the graph convolution. Specifically, SGC [2] reduces the entire procedure of GCN to a simple feature propagation step and their experimental evaluation demonstrates that these simplifications not only do not negatively impact accuracy in many downstream applications but also enable the model to scale to larger datasets and yields up to two orders of magnitude speedup over FastGCN. For specific downstream tasks, i.e. recommender systems, lightGCN [3] observes that feature transformation and nonlinear activation have no contribution to the performance and even increase the training difficulty. On the contrary, removing them leads to significant accuracy improvements and prevents overfitting problem.
>
> 3, **As a matter of fact, our simplified GCN module can be seen as a special “decoupled GCN” [4-6].** Unlike traditional GCN, where the EP and ET operations are stacked together in each convolutional layer of GCN, the decoupled GCNs try to decouple these two operations. Thereinto, one pattern is the Disentangled Propagation and Transformation (DPT), where EP is executed in advance then training the neural network composed of multiple ET operations (also called Propagation then Training as described in [6]). For our method, the simplified GCN module can be seen as the EP operation. After that, the generator (one linear layer) in GAN is acted as the ET operation. Together, they form a decoupled GCN and the structural information is propagating through the entire graph through this decoupled GCN.
>
> We can also see that our GCN module contributes to the performance through the Ablation Study in Table 2. The model performance will consistently drop in all language pairs and even fail to converge in distance language pairs if we further remove the GCN module. We can get the following conclusion: GCN module can capture much richer semantics by extracting structure information of embeddings space, which contributes to learning a better and robust mapping function which stabilizes the BLI performance;
>
> Thanks for the valuable suggestion. For better understanding, we will add “Why  it works” discussion in the revised section, and a more detail experiment will be added to explore the effects of W in the revised manuscript.
>
> [1] Thomas N. Kipf and Max Welling. 2017. Semi-Supervised Classification with Graph Convolutional Networks. In ICLR.
>
> [2] Wu, Felix, et al. "Simplifying graph convolutional networks." International conference on machine learning. PMLR, 2019.
>
> [3] He, Xiangnan, et al. "Lightgcn: Simplifying and powering graph convolution network for recommendation." Proceedings of the 43rd International ACM SIGIR conference on research and development in Information Retrieval. 2020.
>
> [4] Liu, Meng, Hongyang Gao, and Shuiwang Ji. "Towards deeper graph neural networks." Proceedings of the 26th ACM SIGKDD international conference on knowledge discovery & data mining. 2020.
>
> [5] W. Zhang, Z. Sheng, Y. Jiang, Y. Xia, J. Gao, Z. Yang, and B. Cui, “Evaluating deep graph neural networks,” arXiv preprint arXiv:2108.00955, 2021.
>
> [6] Dong, Hande, et al. "On the equivalence of decoupled graph convolution network and label propagation." Proceedings of the Web Conference 2021. 2021.
>
>
> ## Comment3 :
> For experiments, I have 2 comments - (i) addition of performance on word similarity and sentence translation tasks as in the MUSE paper (and others) would lend more credibility to the robustness and effectiveness of the framework. (ii) addition of morphologically rich languages like Finnish, Hebrew, etc and low-resource languages in the experiments would be good to have (minor point).
> ### Response:
> 1, Thanks for suggestions. As suggested by the reviewer, we e added the experiments of word similarity and sentence translation tasks following MUSE paper, as shown in table A and table B.
>
> |TableA: Sord Sim|En-De|	De-En|En-Es|Es-En|
> |---|---|---|---|---|
> |(Lample et al., 2018b) 	|	0.708|	0.713	|	0.712|	0.711|
> |(Artetxe et al., 2018b) 	|	0.719|	0.719|		0.721|	0.721|
> |(Mohiuddin and Joty, 2019) 	|	fail	|0.72|		0.724|	0.718|
> |SA-GAN without PLM	|	0.721|	0.721	|	0.723|	0.723|
> |SA-GAN	|**0.726**	|**0.726**		|**0.725**	|**0.725**|
>
> Table A shows that SAGAN achieves a better Pearson’s correlation to human-annotated scores across languages (except it) – providing better alignment across languages.
>
> |TableB: Sentence Trans|		en-es|	es-en	|	en-fr|	fr-en|
> |---|---|---|---|---|
> |(Lample et al., 2018b)	|	75.1|	73.9|		69.1|	69.9|
> |(Artetxe et al., 2018b)		|74.7	|74.4		|69.6	|69.3|
> |(Mohiuddin and Joty, 2019)		|75.0|	75.7|		68.0	|71.0|
> |SA-GAN without PLM|		73.3|	73.7	|	71.2|	71.8|
> |SA-GAN 	|	**75.4**|	**76.0**	|	**73.7**|	**74.4**|
>
> Table B shows sentence translation retrieval result on Europarl corpus. Table B depicts that SAGAN provides better sentence translation retrieval accuracy, with up to 3.4% score improvements.
>
> 2, To better test our model’s robustness and effectiveness, we added more experiments on morphologically rich languages, i.e., Finnish (Fi), Hebrew (He), and Romanian (Ro) following Oprea [1]. The results ae presented in Table C. From the measurements, we can see that our approach outperforms existing methods on all of six tasks on morphologically rich languages pairs, with a gain up to 1.4% on En-He and 0.8% on average of all languages, which further shows the robustness and effectiveness of our framework.
>
> |TableC: morphologic|		en-fi|	fi-en	|	en-he	|he-en	|	en-ro|ro-en|	avg|
> |---|---|---|---|---|---|---|---|
> |(Lample et al., 2018b)|		43.7|	53.7	|	38	|fail|		58.0|	66.0	|	51.9|
> |(Artetxe et al., 2018b)		|49.9|	63.5| 44.6|57.7	|64.2	|71.8	|	58.6|
> |(Mohiuddin and Joty, 2019)	|	49.8|	65.5	|	46.1|	58.6	|	62.6|	71.9	|	59.1|
> |(Oprea et al., 2022) [1]		|49.9|	65.5|		46.6|	59.1|		65.4|	**74.3**	|	60.1|
> |SA-GAN without PLM	|	52.3|	64.9|		47.1|	57.5|		66.2|	72.3|		60.1|
> |SA-GAN  		|**52.7**	|**66.0**		|**48.0**	|**59.7**	|**66.4**	|72.7		|**60.9**|
>
> [1] Oprea, Silviu Vlad, Sourav Dutta, and Haytham Assem. "Multi-Stage Framework with Refinement Based Point Set Registration for Unsupervised Bi-Lingual Word Alignment." Proceedings of the 29th International Conference on Computational Linguistics. 2022.
>
> ## Comment4 :
> My major question is about the GCN - what does it do, if it does not learn anything? How is, then, the structural information propagated within the graph?
> ### Response:
> We thank the reviewer for the careful review and valuable suggestions. A detail response can be found in the response of the above comment 2.

---

### Meta-Review · Area_Chair_1GXc · 2023-09-08

**Recommendation:** 4

**Metareview:**

The paper proposes a new method for word-embedding-based bilingual lexicon induction (BLI) using generative adversarial networks (GAN). The way GANs are used is taken from previous work. The main innovation is using a graph convolutional network to inform word embeddings about the topological structure of the embeddings, i.e., about its neighbors in the space (although one review disputed the terminology, claiming it is not really GCN).

Overall, the reviewers mainly criticized somewhat weak experimental results and somewhat limited novelty. During the discussion period, the reviewers raised their scores from rather negative to rather positive.

---

### Decision · Program_Chairs · 2023-10-07

**Decision:**

Accept-Findings

**Comment:**

The paper proposes a new method for word-embedding-based bilingual lexicon induction (BLI) using generative adversarial networks (GAN). The way GANs are used is taken from previous work. The main innovation is using a graph convolutional network to inform word embeddings about the topological structure of the embeddings, i.e., about its neighbors in the space (although one review disputed the terminology, claiming it is not really GCN).

Overall, the reviewers mainly criticized somewhat weak experimental results and somewhat limited novelty. During the discussion period, the reviewers raised their scores from rather negative to rather positive.